

# Where are the Penaeids crustins?

Marcel Martinez-Porchas[1], Jorge Hernández-López[2] and Francisco Vargas-Albores[1]

[1] Centro de Investigación en Alimentación y Desarrollo, A.C., Hermosillo, Sonora, Mexico
[2] Centro de Investigaciones Biológicas del Noroeste, Hermosillo, Sonora, Mexico

Corresponding author
Francisco Vargas-Albores,
fvalbores@ciad.mx

## ABSTRACT

Crustins are antimicrobial peptides and members of the four-disulfide core (4-DSC) domain-containing proteins superfamily. To date, crustins have only been reported in crustaceans and possess a structural signature characterized by a single 4-DSC domain and one cysteine-rich region. The high-throughput sequencing technologies have produced vastly valuable genomic information that sometimes dilutes information about previously sequenced molecules. This study aimed (1) to corroborate the loss of valuable descriptive information regarding crustin identification when high throughput sequencing carries out automatic annotation processes and (2) to detect possible crustin sequences reported in Penaeids to attempt a list considering structural similarities, which allows the establishment of phylogenetic relationships based on molecular characteristics. All crustins sequences reported in Penaeids and registered in the databases were obtained. The first list was made with the proteins reported as crustin or carcinin, excluding those that did not meet the structural characteristics. Subsequently, using local alignments, sequences were sought with high similarity even if they had been reported with a different name of crustin but with a probability of being crustin. This broader list, including proteins with high structural similarity, can help establish phylogenetic relationships of shrimp genes and the evolutionary trajectory of this antimicrobial distributed exclusively among crustaceans. Results revealed that in most sequences obtained by Sanger or transcriptomics, which met the structural criteria, the identification was correctly established as crustin. Contrarily, the sequences corresponding to crustins obtained by whole genome sequencing projects were incorrectly classified or not characterized, being momentarily "buried" in the information generated. In addition, the sequences that complied with the criteria of crustin tended to be grouped into species separated by geographical regions; for example, the crustins of the inhabitant shrimp of the American coasts differ from those corresponding to the natives of the Asian coasts. Finally, the results suggest the convenience of annotations considering the previous but correct information, even if such information was generated with previous technologies.

## INTRODUCTION

Crustins are a cysteine-rich hydrophobic antimicrobial protein (AMP), members of the four-disulfide core (4-DSC) domain-containing proteins superfamily. The 4-DSC domain, also named WAP (whey acidic protein), comprises eight cysteine residues involved in

disulfide bonds in a conserved arrangement (*Hennighausen & Sippel, 1982*). Crustins have a molecular weight ranging from 10 to 19 kDa and contain a unique 4-DSC domain at the C-terminus, preceded by one cysteine (Cys)-rich region. A glycine (Gly)-rich motif could be present between the signal peptide and Cys-rich region (*Bartlett et al., 2002*; *Supungul et al., 2004*; *Vargas-Albores et al., 2004*; *Hauton, Brockton & Smith, 2006*; *Smith et al., 2008*; *Tassanakajon, Somboonwiwat & Amparyup, 2015*; *Vargas-Albores & Martínez-Porchas, 2017*). The first crustin, named carcinin, was isolated from the shore crab blood cells *Carcinus maenas* (*Relf et al., 1999*) and characterized as a cysteine-rich 11.5 kDa AMP with antimicrobial activity against Gram-positive or salt-tolerant bacteria (*Relf et al., 1999*; *Brockton, Hammond & Smith, 2007*). Afterward, similar proteins were reported as a virtual translation of sequences from *Penaeus* (*Litopenaeus*) *vannamei* and *P. setiferus* hemocytes cDNA libraries, then the name crustin was proposed (*Bartlett et al., 2002*) because of their distribution among crustaceans. In the same study, six isoforms of *P. vannamei* crustin were reported, differentiable by the number (4–5) of repetitive glycine units and a pair of substitutions outside the functional domains. Since then, crustin has been reported in a variety of crustaceans, including *P. vannamei* (*Vargas-Albores et al., 2004*), *P. monodon* (*Supungul et al., 2004*), *P. (Marsupenaeus) japonicus* (*Rattanachai et al., 2004*), *P. (Fenneropenaeus) chinensis* (*Zhang et al., 2007*), *Panulirus argus* (Stoss et al., 2003), *Homarus gammarus* (*Hauton, Brockton & Smith, 2006*), *C. maenas* (*Brockton, Hammond & Smith, 2007*) and *Pacifastacus leniusculus* (*Jiravanichpaisal et al., 2007*).

Crustins seem to be distributed only in crustaceans, although *Zhang & Zhu (2012)* reported crustin-like sequences in seven ant genomes. However, unlike typical crustacean crustins, an aromatic amino acids-rich region separates the 4-DSC domain and the Cys-rich region, and transcription or antimicrobial activity has not been demonstrated. On the other hand, other proteins related to the immune system and containing one or more 4-DSC domains have also been considered crustins, although they do not have the characteristic Cys-rich region. Therefore, it is necessary to establish solid classification criteria to improve understanding of the evolution, biological role, and importance of this AMPs group, avoiding conceptual discrepancies preventing adequate evolutive and phylogenetic association while maintaining biological coherence in this type of molecule.

With the advent of massive sequencing, many sequences were either revealed, completed, or reaffirmed. However, this brought a need for strict scrutiny to make the corresponding annotations considering that the annotated sequences belong to molecules with proven activity. In addition, this scrutiny is required to avoid the loss of information due to the substitution of previous sequences by new ones with other annotations, not considering former but essential information about the molecule. For instance, whole genome sequencing (WGS) and RNA sequencing (RNAseq) projects in penaeids have been carried out, reporting dozens of sequences with the 4-DSC domain in four species, *P. vannamei*, *P. monodon*, *P. japonicus*, and *P. chinensis*; however, very few sequences have been reported or annotated as crustins, which is an unexpected outcome considering the importance that crustacean immunologists have conferred to these molecules. In the first instance, this could be evidence of a loss of relevant information from important molecules when capturing information derived from new technologies in databases, possibly representing a setback

in the correct annotation of this type of molecule. For this reason, this study aimed (1) to corroborate the loss of valuable descriptive information regarding crustin identification when high throughput sequencing carries out automatic annotation processes and (2) to detect possible crustin sequences reported in Penaeids to attempt a list considering structural similarities, which allows the establishment of phylogenetic relationships based on molecular characteristics.

## MATERIALS AND METHODS

### Abreviatures

The scientific names of the Penaeids species are abbreviated as five-letter capital labels; the first three belong to the genus, and the last two to the species. PENBR (*P. brasiliensis*); PENCH (*P. chinensis*); PENIN (*P. indicus*); PENJA (*P. japonicus*); PENMN (*P. monoceros*); PENMO (*P. monodon*); PENPA (*P. paulensis*); PENPE (*P. penicillatus*); PENSC (*P. schmitti*); PENSE (*P. setiferus*); PENSU (*P. subtilis*); PENVA (*P. vannamei*).

### Data collection

Sequences were firstly searched in GenBank non-redundant protein database and, therefore, in the Identical Proteins Groups database using the terms "crustin OR carcinin". A reference sequence (RefSeq) is defined as a non-redundant and richly annotated sequence representing a naturally occurring DNA, RNA, or protein molecule (*Pruitt et al., 2002, 2012*), in case of duplicates, the RefSeq was selected as representative. Sequences fully contained in another sequence were also excluded. Because the Cys content is essential to maintain the characteristic Cys arrangement of the crustin signature (Cys-rich region + 4-DSC domain), only sequences with 12 Cys and at least two CysCys (CC) pairs were considered.

### BlastP

The BlastP algorithm (*Camacho et al., 2009*), implemented at the National Center for Biotechnology Information (NCBI: https://blast.ncbi.nlm.nih.gov/Blast.cgi), was used to establish similarity by local alignment in a GenBank non-redundant protein database. The maximum required target sequences was 250 hits, with word size = 5, expected threshold 0 0.05, and matrix BLOSUM62 were used. Considering that an E value $<E-50$ indicates an almost identical sequence (*Altschul et al., 1990*; *Scholz, 2020*), we decided to reduce the strictness and use an E value $<e-20$ to select only closely related sequences and maintain homogeneity, counting that they share a domain.

### Bioinformatics analysis

The signal peptide and the secretory property were determined by the signalP program (https://services.healthtech.dtu.dk/services/SignalP-5.0/) (*Nielsen et al., 2019*). Clustal O (*Sievers et al., 2011*), as implemented in the European Bioinformatics Institute (https://www.ebi.ac.uk/Tools/msa/clustalo/), was used for multiple alignments. The phylogenetic trees were inferred by Maximum parsimony (bootstrap = 1000) using the MEGA 11 program (*Tamura, Stecher & Kumar, 2021*).

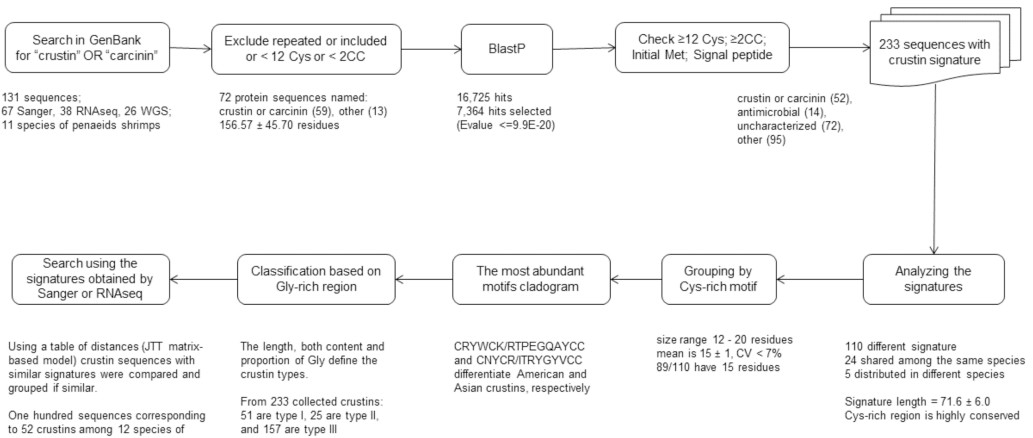

**Figure 1** Scheme to obtain sequences meeting the criteria established to be considered as crustins. The pipeline was as follows: 1. Crustin or carcinin sequences were searched in the GenBank; 2. repeated or incompleted sequences and those containing <12 Cys or <2 CC pairs were excluded; 3. additional sequences similar to any listed crustin were searched by BlastP, and those having an E value <=9.9E−20 were selected; 4. sequences without signal peptide or initial Met or containing <12 Cys or <2 CC pairs are considered incomplete and excluded; 5. only 233 sequences having the crustin signature (Cys-rich region and 4-DSC domain) were selected; 6. crustin signatures were used as the first classification criterium; 7. grouped by their Cys-rich motif, the most conserved part of the signature; 8. cladograms with the most abundant motifs were done; 9. crustin types were assigned based on the Gly-rich region; and 10. a table of distances was used for comparison, and group crustins with similar signatures.

## Statistical analysis

One-way ANOVA and Tukey test (SPSS, IBM Corporation, USA) were used to evaluate significant differences ($P < 0.05$) for some measurement as sequence size or Gly content.

## RESULTS & DISCUSSION

Overall, a ten-step bioinformatics pipeline was performed to obtain crustin sequences that met the criteria established for these kinds of antimicrobial molecules, including conserved sequence features such as cysteine content, initial and signal peptides, molecule signature, motifs, etcetera (Fig. 1).

### Data collection

Since the first description of carcinin, later called crustin, many other sequences have been reported. In the initial search, 219 protein sequences matching the words crustin or carcinin from Crustaceans species were retrieved from the GenBank non-redundant protein database, from which 108 belonged to Penaeid species, suggesting that crustins are exclusively distributed in crustaceans. Crustin sequences were also found in the Identical Proteins Groups database, and 23 new records were retrieved. Many of these proteins were not detected in the first search because they were registered under a different name. Even more, 19 of the 23 new records showing identical sequences to crustins are considered RefSeq (*Pruitt et al., 2002, 2012*), but surprisingly none is named or annotated as crustin or carcinin. Some current names are related to any antimicrobial activity (putative

antimicrobial peptide, antileukoproteinase-like, elastin-like, CruU). In contrast, others refer to proteins not directly related to the defense system (ATP-dependent RNA helicase, calcium-binding protein P, epsin) or are even defined as uncharacterized or hypothetical proteins. These 23 sequences and the 108 sequences named crustin or carcinin (131 in total) all belonged to 11 species of penaeid shrimp (Table S1). Of these, 67 sequences were revealed by Sanger sequencing, 38 by RNAseq, and 26 resulted as read assemblies performed in WGS projects.

## First selection

Next step, replicated sequences were excluded, leaving one representative, the RefSeq, if it exists. Similarly, sequences included in another more extended sequence were excluded, reducing the listed sequences from 131 to 93. The proportion of sequences achieved by WGS increased because they were retained as they are considered RefSeqs. The sequences from WGS are slightly longer ($167.95 \pm 97.43$) than those obtained by RNAseq ($144.28 \pm 53.53$) or Sanger ($149.89 \pm 4.80$); however, ANOVA-1W reported no significant differences. Considering the requirement of 12 Cys and 2 CC pairs for the molecular signature, sequences not meeting this requirement or having any additional domain, such as secretory leukocyte proteinase inhibitor (SLPI) or DWD-proteins (*Jiménez-Vega & Vargas-Albores, 2007*; *Chen, He & Xu, 2008*; *Du et al., 2009*), were also excluded. Thus, screening the characteristic molecular arrangement of the crustin signature resulted in excluding 19 sequences, even when some were reported as crustin, and some other proteins without antimicrobial activity (beta-actin, ATP-dependent RNA helicase A-like, epsin-like). The remaining 72 protein sequences, sizing $156.57 \pm 45.70$ residues, are recorded as crustin (58), carcinin (one), and antimicrobial (one) as well as known proteins, including WAP four-disulfide core domain containing-like, small cysteine, and glycine repeat-containing protein 2, calcium-binding protein P, antileukoproteinase, elastin, cuticle protein 64, and 41 kDa spicule matrix protein. The rest were only assigned as uncharacterized proteins (Table S1).

## BlastP

Although with different names, all 72 sequences have the characteristic crustin molecular signature, Cys-rich region, and 4-DSC domain, sizing $69.48 \pm 7.79$ residues. The modest coefficient of variation (CV $=11.02$) indicates that the signature size is conserved, and its influence on variations in the total protein size is less than in other regions. Substantial evidence indicates that proteins that could be characterized as crustins by meeting the molecular criteria are currently annotated under other names. Thus new sequences, similar to any listed crustin, were searched in the GenBank non-redundant protein database by BlastP. Although varying the hit number for each sequence, 16,725 hits were retrieved; however, only 7,364 hits with E value $<=9.9E{-}20$ were selected. Additionally, sequences with multiple hits were not considered since crustin does not have repeat domains. After that, 24 crustins were recurring, and over 200 hits (with E value $<=9.9E{-}20$) were detected, while 41 crustins barely reached 50 hits, and seven crustins got between 50 and 200 hits (Table S2). This difference was used for selecting sequences with the highest probability of being crustin or the most similar molecular structure.

The above 7,364 hits with E value <=9.9E−20 correspond to 404 different sequences; however, only 204 were considered new proteins because of any of the following reasons: (1) some were already included in the original list, (2) have both low similarity and a low number of queries, (3) have two crustin signatures or have less than 12 Cys. In this set of new proteins, 16 were associated with the 4-DSC domain (antileukoproteinase, elastin, whey acidic protein-like) or registered as an antimicrobial peptide. However, most were classified as other types of protein, including 77 (36.6%) uncharacterized proteins, 14 (6.7%) trihydrophobin-like, 14 (6.7%) glycine-rich cell wall structural proteins, 11 (5.24%) pupal cuticle protein, and 11 (5.24%) ctenidin-1-like protein. Because crustin is a secreted protein, the signal peptide and the initial Met were searched in the newfound sequences. Neither of these characteristics was found in two sequences (XP_047502908, XP_037772699) obtained by WGS and in five sequences obtained by Sanger (AAS57715, ADF80918, ACZ43781, QHD40387, ACT82964), indicating that such sequences are truncated. In addition, 19 sequences from WGS projects were excluded because, despite the initial Met, the signal peptide cleavage site was not detected. Finally, BlastP reported similarity with 17 sequences from no penaeid crustaceans, including *Homarus americanus, Hyalella azteca, Macrobrachium nipponense, M. rosenbergii, Neocaridina davidi, Pandalus japonicus, Panulirus japonicus, Portunus trituberculatus*, and *Trinorchestia longiramus*. These sequences were also excluded in the subsequent analyses focused on shrimp crustins.

In this way, a collection of 233 sequences of crustins and proteins with a similar sequence was reached; all of them should have the characteristic crustin molecular signature, although optionally, they could have the Gly-rich region between the Cys-rich region and the signal peptide. The global alignment by Clustal O did not offer a clear resolution due to the great diversity of sizes and composition, especially from regions outside the crustin signature, which is the most conserved region. Each sequence, and its main characteristics, are listed in Table S3.

In the 72 sequences listed before BlastP (Fig. 2), only nine (12.5%) sequences were reported as "uncharacterized" or another protein unrelated to antimicrobial activity. However, this proportion increased enormously (71.7%) after retrieving similar sequences by BlastP due to incorporating products obtained from WGS projects of PENVA, PENMO, PENJA, and PENCH. Consequently, the participation of crustin or antimicrobial sequences was reduced to 28.3%, indicating the loss of information regarding annotations for proteins that meet all the criteria to be recognized as crustin when recorded from WGS projects. This is possible because many annotations are carried out automatically without incorporating pre-existing information from the same proteins registered with previous technologies, which is a problem caused by the vast differential between the information generated with new technologies and the capacity of annotation; thus, the annotations should be automatized for considering the preexistent information.

## The shrimp crustin signature

The size of the 233 sequences containing the crustin signature (179.0 ± 53.3) showed a high CV (29.8%), while the size of the signatures (72.2 ± 4.5) showed less CV (6.2%), indicating that the signature is a feature highly conserved and has less influence on the size
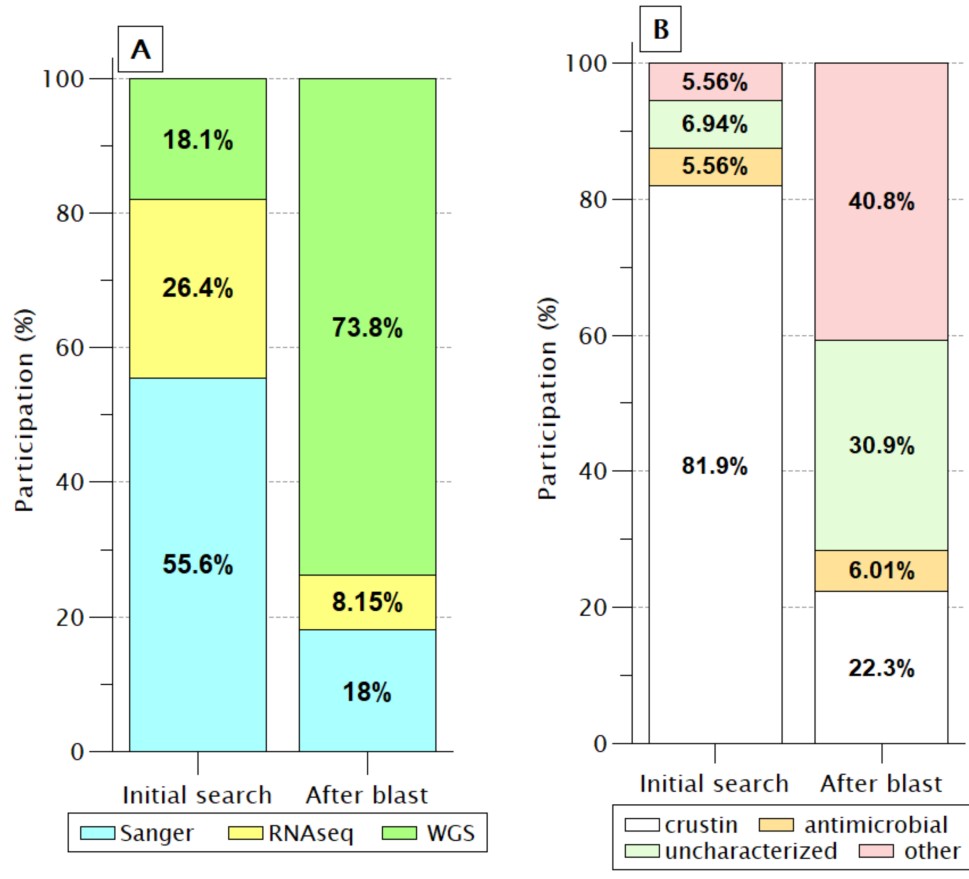

**Figure 2** **Identification labels of molecules meeting the crustin criteria.** (A) Percentages are presented, according to the sequencing technique; Without a doubt, WGS has provided the most significant proportion. (B) As the participation of sequences produced by WGS increases, the proportion of sequences reported as crustin decreases, while the proportion of those reported as uncharacterized or other proteins increases.

than other protein's regions. Although the sequence and signatures obtained by WGS are slightly larger (185.3 ± 54.3 & 73.2 ± 2.5) than those obtained by Sanger (163.8 ± 42.3 & 68.4 ± 6.8) or RNAseq (155 ± 52.2 & 72.4 ± 6.7), the differences do not seem to be significant (Fig. 3). The crustin signatures were arbitrarily numbered only for identification purposes; the length, organism source, and using technology for each one are shown in Table S4. Although most signatures (81) are exclusive to a unique sequence, 29 signatures are distributed in multiple sequences, mainly in species with abundant reported sequences (PENCH, PENJA, PENMO, and PENVA). Thus, the 233 sequences are covered with 110 different signatures, reflecting its conservative character. The signature is also shared among sequences from different species (Table 1); the extreme example is the signature number 176, repeated in 36 sequences from three *Penaeus* species.

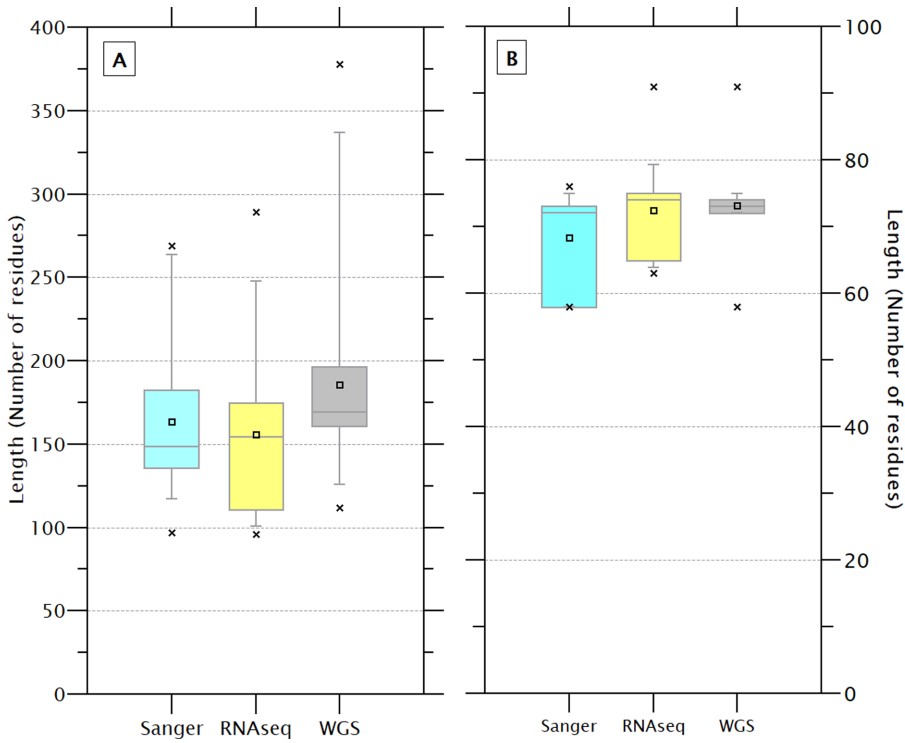

**Figure 3** **The length of the complete sequence (A) and crustin signature (B) of the 233 selected sequences varies slightly depending on the technique used for sequencing.** These differences are no significant. A higher dispersion is observed in the sequences obtained by WGS than Sanger.

**Table 1** **Distribution of crustin signatures.** Crustin signatures can be found in different proteins from the same species and in crustins from different species. Most cases are observed in species whose genome has been sequenced.

| Signature | Shrimp | Freq |
|---|---|---|
| 008 | *Penaeus paulensis* | 1 |
| | *Penaeus subtilis* | 1 |
| 137 | *Penaeus chinensis* | 1 |
| | *Penaeus monodon* | 4 |
| 108 | *Penaeus setiferus* | 1 |
| | *Penaeus vannamei* | 4 |
| 132 | *Penaeus chinensis* | 5 |
| | *Penaeus monodon* | 2 |
| | *Penaeus vannamei* | 3 |
| 176 | *Penaeus chinensis* | 8 |
| | *Penaeus monodon* | 10 |
| | *Penaeus vannamei* | 18 |

## The Cys-rich region

The molecular signatures found in penaeids crustins have a length of $71.6 \pm 6.0$ residues (CV = 8.4); the significant size variation is in the interdomain ranging from four to 18

**Table 2 Crustins from penaeid species.** The number of crustins and their types correspond to 12 penaeid species detected in databases.

| Species | Crustin type | | | Total per species |
|---|---|---|---|---|
| | I | II | III | |
| *Penaeus brasiliensis* | 0 | 1 | 0 | 1 |
| *Penaeus chinensis* | 3 | 0 | 1 | 4 |
| *Penaeus indicus* | 0 | 0 | 1 | 1 |
| *Penaeus japonicus* | 3 | 2 | 2 | 7 |
| *Penaeus monoceros* | 0 | 0 | 1 | 1 |
| *Penaeus monodon* | 0 | 1 | 5 | 6 |
| *Penaeus paulensis* | 0 | 1 | 0 | 1 |
| *Penaeus penicillatus* | 1 | 0 | 1 | 2 |
| *Penaeus schmitti* | 0 | 1 | 0 | 1 |
| *Penaeus setiferus* | 1 | 1 | 1 | 3 |
| *Penaeus subtilis* | 0 | 1 | 0 | 1 |
| *Penaeus vannamei* | 13 | 3 | 8 | 24 |
| Totals | **21** | **11** | **20** | **52** |

amino acids, while the Cys-rich region seems to be the most conserved part of the signature. Although the Cys-rich region size range is 12–20 residues, the mean is $15 \pm 1$ (CV <7%), where 89 out of 110 fragments have 15 residues. Furthermore, some Cys-rich regions are shared among signatures and, consequently, can be located in multiple whole protein sequences (Table S5). The most frequent Cys-rich sequence was CRYWCKTPEGQAYCC which was detected in 33 signatures, corresponding to 92 (34.5%) crustin sequences. The wide distribution of this Cys-rich motif is also appreciated by its location in nine recorded species of penaeids, thus being the most representative motif for the crustin penaeids (Table S6). Nine of these 92 CRYWCKTPEGQAYCC-containig signatures were obtained by the Sanger method and belong to six shrimp species, while only one produced by RNAseq belonged to PENVA. The remaining 82 signatures correspond to sequences generated through genome sequencing of PENVA, PENMO, PENCH, and PENJA.

Crustin signatures containing this motif were aligned, and the phylogenetic relationship was analyzed by Maximum parsimony, implemented in Mega 11. One clade contains five signatures found in ten sequences, obtained by Sanger, from six penaeid species (PENBR, PENPA, PENSC, PENSE, PENSU, PENVA) exclusively distributed on American coasts and reported as crustin or antimicrobial peptides. The condensed tree, collapsing sequences from the same species, is shown in Fig. 4, while the uncondensed tree and other cladograms are in Table S7. The most significant difference between the sequences is in the Gly-rich region, whose characteristics are described and discussed later. A ten-signature group from PENJA includes 32 sequences obtained by WGS and reported as trihydrophobin, pupal cuticle protein 20, keratin-associated protein, uncharacterized protein, or other not related to the immune system proteins. Similarly, one group, including sequences from PENMO (24), PENCH (15), and PENVA (three), together with others containing sequences from PENMO (five) and PENCH (two) have and share the crustin signatures. However, they

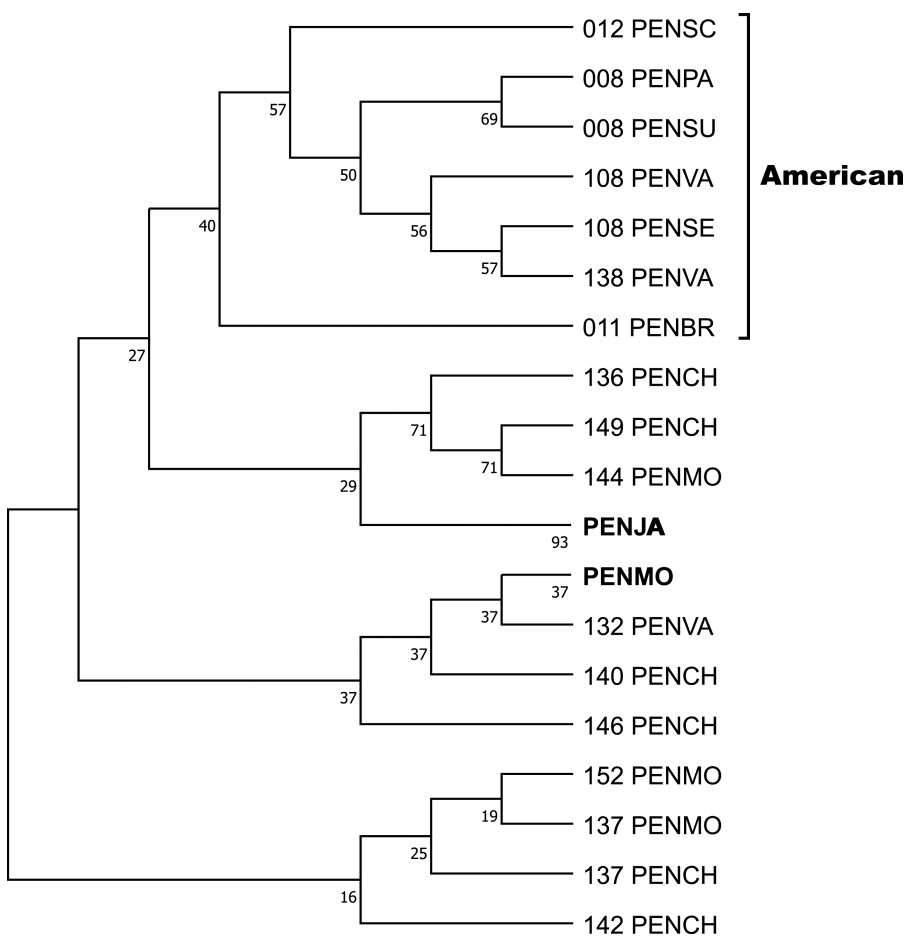

**Figure 4** **The phylogenetic relationship of the signatures containing the Cys-rich domain CRYWCK-TPEGQAYCC was inferred using the Maximum Parsimony method, and the bootstrap consensus tree inferred from 1,000 replicates is shown.** The sequences of six species of penaeids distributed along the American coasts are located in a clade. Sequences from *P. japonicus* and *P. monodon* were pooled. The non-condensed tree is found in Table S5.

were reported as trihydrophobin, chaperone, or uncharacterized protein, including its isoforms, even from the same locus, but not as crustin or antimicrobial.

Differing only by one conservative replacement (K/R6), the second most frequent Cys-rich region is CRYWCRTPEGQAYCC found in 13 signatures corresponding to 51 sequences of PENCH, PENJA, PENMO, and PENVA, species whose genome has been described and where the highest number of signature repetitions is observed. This group does not include sequences obtained by Sanger nor proteins named crustin, except for QOL09958, obtained by RNAseq (Table S7). The signature 176 (arbitrarily numbered) is located in 36 sequences from PENCH, PENMO, and PENVA registered as ctenidin, glycine-rich cell wall structural protein or uncharacterized protein, including 15 isoforms from PENVA LOC113802945.

Thus, 143 sequences (61.4%) and 39 signatures (35.4%) from nine (75%) penaeid species contain the consensus motif CRYWC(K/R)TPEGQAYCC in the Cys-rich region. This motif was found by BlastP only in the other five cases. One is the PENMO uncharacterized protein LOC119568318; however, it does not have the characteristic 4-DSC domain (XP_037772700). Although with different names, the other four sequences have the full crustin signature but are not penaeids: the hypothetical protein (KAA0200687) and the glycine-rich cell wall structural protein (XP_018011805) from *Hyalella Azteca*, the *Trinorchestia longiramus* WAP-type 'four-disulfide core' domain (KAF2367912) and the *Macrobrachium nipponense* glycine-rich crustin 1 (QHG61850). Although this motif was not found among the dozens of species where crustins have been reported, its distribution appears to be only in crustaceans and possibly only in crustins.

Interestingly the third abundant Cys-rich region CNYCRTRYGYVCC or its isoform CNYCITRYGYVCC was located in another geographically distributed group of shrimp. Unlike the more abundant 15-residue motif (CRYWC(K/R)TPEGQAYCC), this consensus (CNYC(R/I)TRYGYVCC) has only 13 residues and is located in nine highly similar PENMO signatures and one each from PENIN, PENPE, PENCH, and PENMN. All sequences were obtained by Sanger and reported as crustins or antimicrobial peptides, except the PENCH uncharacterized protein (XP_047502940), although it is identical to the crustins AAZ76017 and QOL09965, obtained by Sanger and RNAseq, respectively. Thus, the motif CNYC(R/I)TRYGYVCC could be representative of Asian shrimp crustins as long as it is not reported in American shrimp. Besides demonstrating that Sanger sequencing provided a confident and consistent crustin characterization, these results provide a coherent crustin differentiation between different penaeid species separated by a significant geographical space.

Recapping, two consensuses motifs, CRYWC(K/R)TPEGQAYCC and CNYC(R/I)TRYGY, were located in 50 of 110 (45.5%) signatures and in 156 of 233 (70%) sequences belonging to all shrimp species where they have been reported. However, few sequences are currently named crustin; instead, most have been recorded as uncharacterized (35.9%) or as other proteins (48.1%), and there are no elements to identify them as crustins. Only 25 (16%) sequences (21 by Sanger sequencing) were recorded as crustin or antimicrobial peptides. In terms of taxonomic distribution, sequences from American shrimp have a very different signature from that found in Asian shrimp: (a) the Cys-rich size is 15 in Americans and 13 in Asians, (b) while the interdomain is 16 in Americans, while in Asians it is remarkable as short as four residues; (c) therefore, the signature of the crustins is longer in American shrimp than in Asian ones; 72 and 58 residues, respectively, once again demonstrating a clear differentiation between taxonomic groups (Fig. 5).

## The Gly-rich region

The Gly-rich region is another crustins structural feature located between the signal peptide and the crustin signature and has been used as a classification criterion for crustins (*Smith et al., 2008*; *Vargas-Albores & Martínez-Porchas, 2017*). Most classification schemes consider that type I crustins do not have the Gly-rich region, or such region is short. One proposal (*Vargas-Albores & Martínez-Porchas, 2017*) considers the Gly percent for

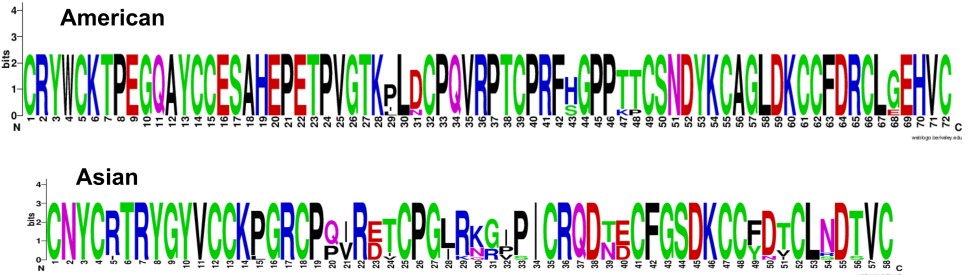

**Figure 5 Logo of crustin signatures.** Different crustin signatures in shrimp distributed along the American and Asian coasts.

the crustins classification: $50.7 \pm 8.4\%$ for type II and $28.5 \pm 10.1$ for type III. Thus, the Gly-rich region does not seem very relevant for antimicrobial activity since some type I crustins do not have this fragment, while in types II and III, this region is very diverse in length and content. The extent of the Gly-rich region and its content and proportion of the American and Asian shrimp were compared. As shown in Fig. 6, rather than the size or the Gly content, the proportion of this amino acid seems to be the best criterion for the classification of type II and III crustins. According to the Gly percent, the American ($57 \pm 2.36\%$) and Asian ($38.68 \pm 3.18\%$) crustins are type II and III, respectively. Under this criterion, the 233 crustin or structurally similar sequences in the GenBank are distributed as follows: 51 are type I, 25 are type II, and 157 are type III. However, only 25 reported crustin-like sequences were found using the most abundant Cys-rich domains, so there are still 208 crustin-like sequences that need to be reviewed. Up to this point, the evidence strongly suggests that crustins, sequenced within the genome but not characterized and identified, are momentarily "buried" in the generated information pile.

## Signature as the main characteristics

Both Sanger and RNAseq techniques report transcribed sequences because they analyze mRNA, but the Sanger technique demands much less computing resources for the assembly, and the probability of forming chimeras is practically null. On the other hand, the differences in the Gly-rich region may be due to different isoforms rather than different proteins; therefore, comparing Gly-rich regions is not essential for crustin characterizations but is somewhat problematic sometimes. On the other hand, the crustin signature is highly conserved, so it can be considered the main feature and valuable to find other sequences not described as crustin or antibacterial. Thus, the crustin signatures obtained by Sanger or RNAseq were used for locating similar signatures, according to a table of distances generated using the JTT matrix-based model (*Jones, Taylor & Thornton, 1992*), found in Table S8. The complete sequences with similar signatures were globally aligned(Clustal O) and grouped if they showed considerable similarities, as the same or isoforms. Using this strategy, sequences with CRYWC(K/R)TPEGQAYCC-containing signatures (Table S7) were analyzed. In this group, although with the same signature, the crustins ABO93323 from PENSU and ABM63361 from PENPA have significant differences in other parts of the protein. Similarly, a signature is shared by five PENVA sequences, probably isoforms,
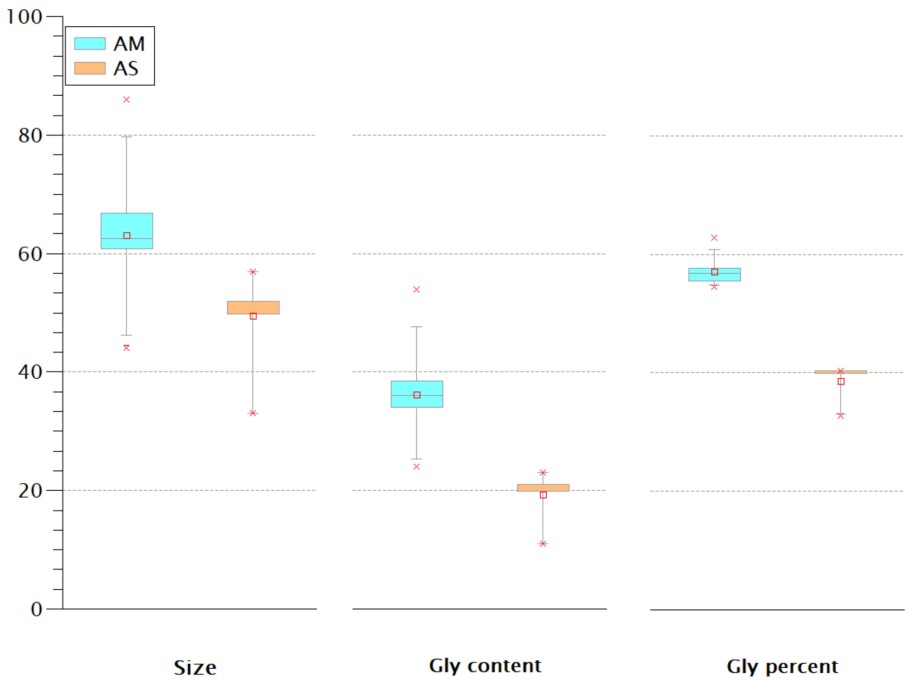

**Figure 6** **The Gly-rich region of American (AM) and Asian (AS) shrimp crustin have different Gly content and percentage.** The percentage may be the best criterion for classifying type II and type III crustins.

because they have variations only in the Gly-rich region. The same signature is present in a PENSE sequence (AAL36897) but with differences in the rest of the molecule. No similar signatures or sequences were detected for the other American shrimp crustin: ABQ96197 from PENBR and ABM63362 from PENSC. Some similarities among the signatures and sequences of the American crustins indicate the conserved character, but there are also typical diversification differences. A sequence produced by PENVA RNAseq was also located that differs from the previous ones and, according to its Gly-rich region, belongs to type III.

On the other hand, with the motif CNYC(R/I)TRYGYVCC, Asian crustins were located, all of them type III, previously detected, and there are no new similar signatures. Six PENMO sequences (AGA83302, ACL97378, ACL97376, ACL97377, ACL97375, ACL97374), reported as a crustin-like antimicrobial peptide, form a group of similar proteins, although with some differences in signature. Another PENMO sequence pair (ACT82963, ADV17347) are very similar, with only a few differences.

However, there are still 77 sequences that have not been analyzed because their signatures do not contain the CRYWC(K/R)TPEGQAYCC or CNYC(R/I)TRYGYVCC motif in the Cys-rich domain. In this group, there are 41 sequences reported as crustin or antimicrobial: 21 obtained by Sanger, 19 by RNAseq, and one by WGS. Reviewing the 21 crustins obtained by Sanger, 14 different signatures were observed, which served to locate two other very similar signatures in proteins not reported as crustins. The 16 signatures of 27 sequences were grouped by type for alignment and visualization. Seven type I sequences were detected;

two originally registered as crustin, ATU82299 (Sanger) and QOL09952 (RNAseq), are almost identical with only two substitutions (distance = 0.0165). The other five sequences did not find similar sequences. In the twelve type II sequences, five PENJA crustins have the same signature and are highly similar, with slight differences in the Gly-rich region. Moreover, these sequences are highly similar to XP_042858712, obtained by WGS and declared as keratin, type II cytoskeletal 3-like; the similarity level suggests that such sequence could be renamed as crustin. Similarly, the XP_042865729 sequence obtained by WGS and described as a probable glycoprotein hormone G-protein coupled receptor only has a few differences with the BBC42585 sequence, described as crustin in PENJA. Finally, in type II sequences, there is a group of four PENMO crustins obtained by Sanger and reported as isoforms, with differences in the Gly-rich region but highly similar in the rest of the molecule. In type III sequences, there is a couple of sequences obtained by Sanger with the same signature and slight variation in the Gly-rich region, which suggests that it is the same protein or isoforms. Furthermore, they are highly similar to a sequence obtained by WGS (XP_037772832) and reported as an uncharacterized protein but could be renamed as crustin. Similarly, the PENJA sequence XP_042865726 reported as a small cysteine and glycine repeat-containing protein 2-like and XP_042865726 as a probable glycoprotein hormone G-protein coupled receptor, could be renamed due to their high similarity to the crustin BBC42584 and BBD52151, respectively.

The same occurs with the sequences produced by RNAseq. The crustin QOL09943, obtained by RNAseq, is very similar to three sequences obtained by WGS and reported as uncharacterized proteins; two are almost identical (XP_027208059, XP_027208058) and were reported as isoforms. Due to the degree of similarity and the absence of the Gly-rich region, all of them could be considered type I crustins. No similar sequences were found for the other, obtained by RNAseq, eight type I crustins. On the other hand, only two type II crustins have been obtained by RNAseq, for which no other similar sequences were found. However, there are five type III PENVA sequences obtained by RNAseq, two with no similarity to any other sequence. The crustins QOL09961 and QOL09960 are similar to XP_027227947 (41 kDa spicule matrix protein-like) and XP_027213162 (cuticle protein 64-like), respectively, with a couple of differences in the Gly-rich region. Observing slight differences in the Gly-rich region, the crustin sequence QOL09955 is highly similar to XP_027208066 (DEAD-box ATP-dependent RNA helicase 52A-like) and to XP_027208065(ctenidin-1-like).

Following the same procedure for the sequences produced by WGS and not reported as crustin or crustin-like is risky due to the lack of transcription evidence. However, those sequences reported as members of the 4-DSC domain-containing protein superfamily could be considered possible crustins. This is the case of XP_027219667 and XP_027224107, reported as antileukoproteinase-like and elastin-like, respectively; however, they do not have the characteristic structure of these proteins; instead, they do have the characteristic crustin signature. As additional evidence, there is a pair of identical sequences produced by WGS, XP_027208057, and ROT82064; the first was reported as an uncharacterized protein, but the second as an antimicrobial peptide. Thus, these four proteins are incorporated as probable crustins.

Finally, several sequences were excluded because they were identical to others and to avoid duplication in the analyses. ABV25095 and ABV25094 are identical to crustin ADV17347, same as QOL09956, AAS59739, AAS59738, AAS59737, AAS59736, AAS59734, AAL36890, AAL36893 are equal to XP_027208055. They are not new crustins but new identical members for the same protein. Many crustin sequences produced by Sanger or RNAseq were masked by an identical sequence produced by WGS but without incorporating pre-existing and valid information provided by Sanger and RNAseq. In this sense, information about identifying this protein type has been lost. Thus, identical sequences were grouped as crustin because at least one of the sequences was reported as such. For example, the sequence XP_047478805 reported as an uncharacterized protein is identical to two sequences obtained by Sanger (ACZ43783, ACD11038) and one by RNAseq (QOL09977) reported as crustin and seven crustins were added to the list. The complete list of penaeid crustins is in Table S9; the summary of crustin by species is shown in Table 2.

Establishing the similarities of the signatures, and even better, that of the Cys-rich region, was a success that allowed us to establish the similarity between American shrimp crustins and differentiate it from Asian shrimp crustins. In that same procedure, it was possible to mark some different characteristics, as expected in evolutionary diversity. Thus, more than 100 sequences corresponding to 52 crustins among 12 species of penaeids were found. However, many sequences produced by WGS have the crustin signature but were reported as other proteins. Undoubtedly, the WGS projects have been of great help in describing new sequences and giving genomic information beyond those transcribed. The diversity outside the crustin signature appears to be higher and phylogenetic relationships should be carefully established due to the presence of multiple isoforms, but it is necessary to advance in confirming its transcription and antibacterial activity.

## ACKNOWLEDGEMENTS

Special thanks to Azucena Santracruz for her contribution to the manuscript format.

### Funding
This work was supported by the Consejo Nacional de Ciencia y Tecnología (CONACyT), Mexico, grant 84398 to Francisco Vargas-Albores. The funders had no role in study design, data collection and analysis, decision to publish, or preparation of the manuscript.

### Grant Disclosures
The following grant information was disclosed by the authors:
Consejo Nacional de Ciencia y Tecnología (CONACyT), Mexico: 84398.

### Competing Interests
The authors declare there are no competing interests.

## Author Contributions

- Marcel Martinez-Porchas conceived and designed the experiments, prepared figures and/or tables, authored or reviewed drafts of the article, and approved the final draft.
- Jorge Hernández-López performed the experiments, analyzed the data, authored or reviewed drafts of the article, and approved the final draft.
- Francisco Vargas-Albores conceived and designed the experiments, analyzed the data, prepared figures and/or tables, authored or reviewed drafts of the article, and approved the final draft.

## Data Availability

The 23 sequences and the 108 sequences named crustin or carcinin (131 in total), all belonging to 11 species of penaeid shrimp; 233 crustins and proteins; crustin signatures; the sizes and sequences of the crustin signature regions; and the list of sequences reported in penaeid shrimp; are available in the Supplemental Files.

## Supplemental Information

Supplemental information for this article can be found online at http://dx.doi.org/10.7717/peerj.15596#supplemental-information.

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
