# Peer review of "Where are the Penaeids crustins?"

_PeerJ, doi:10.7717/peerj.15596_

## Round 0.1 · original submission · Major Revisions

Kindly address each reviewer's comments and concerns with a point-to-point rebuttal.

Reviewer 1 ·

Basic reporting

This study identified sequences of crustins and potential crustins that have been found previously. The effort to improve protein annotations is essential for future research. However, this manuscript needs a major revision as the current state is unclear and unorganised. Full comments are listed below.

Experimental design

The aim of the study was not properly addressed. It has to be consistent in the abstract and introduction.
The methods used were written in an unorganised manner.

Validity of the findings

The findings require a better organisation, maybe, need to be divided into several parts.

Additional comments

1. Please double-check the aim in the abstract and introduction
2. A lot of abbreviations were not properly introduced, e.g., RNA SEQ (in abstract), AMP, CC, CV, nr-database, PENMO, PENVA, PENJA, PENCH etc.
3. Please be consistent in writing particular words, e.g., MEGA 11/ Mega IX, BLAST P/ BLASTP, non-redundant/ nr, RNA SEQ/ RNAseq, Clustal O/ CLUSTAL/ CLUSTAW, etc.
4. It is better to summarise the approaches used in this study, maybe in the last paragraph of introduction.
5. Materials and methods part is not unclear. 2.1 and 2.3 seem redundant. It is better to separate the bioinformatics analysis into parts.
6. The authors need to clearly mention the parameters used for BLASTP, sequence alignment and phylogenetic trees in the methods and justify or provide reference(s) for each parameter used, for example, Evalue < = 9.9E-20.
7. It is preferable to start with a section on data collection in the materials and methods section.
8. Results & Discussion section can be divided into parts.
9. It is better to present a graphical representation of the overall number of crustins, potential crustins, and entries eliminated according to specific criteria.
10. Briefly define condensed and uncondensed trees.
11. Suggestion: Replace signature name instead of ID (Table 1)
12. Add legend for Figure 2.
13. Define AM and AS for Figure 5.

Cite this review as

Reviewer 2 ·

Basic reporting

The author collected crustin sequences of shrimp mainly based on crustin features and blasted them to databases of different sequencing methods to identify homologous sequences with crustin. They considered sequences without names or named as other proteins but with crustin features as crustin-related proteins. Then, they compared the characteristics of crustin genes obtained from different sequencing methods and found that whole-genome sequencing (WGS) method is more efficient in discovering new sequences.
Line 29 “local alignment” should “local alignments”
Line 39 “Asian coast” should “Asian coasts”
Line 112 “OR” should “or”?
Line 118-119 What does the mean “considered reference sequences”? the reference sequences referred to this research or other studies? And how many seuqences in the RefSeq? But these sequences were not named as crustins. I think this sentence is ambiguous.
Line 127 There are not clear. Please clarify the correlation with the 113 sequences?
Line 195 What do the mean of “crustin signature” and “signatures” in the context?
Line 202 Why 233 sequences are covered with 110 different signatures, indicating a high conservation level?
Line 223 Crustin domains or crustin sequences were used to the phylogenetic tree?
Results and Discussions This section contains many methods in this research. The methods should supplement in the Materials and Methods.

Experimental design

no comment

Validity of the findings

no comment

Additional comments

no comment

Cite this review as

---

## Round 0.2 · accepted · Accept

The revised version of the manuscript is now suitable for publication.

Reviewer 1 ·

Basic reporting

All comments have been addressed by the authors.

Experimental design

The methods section has been improved.

Validity of the findings

The results and discussion section has been improved.

Cite this review as